# Applications of Stem Cell-Derived Extracellular Vesicles in Nerve Regeneration

**DOI:** 10.3390/ijms25115863

**Published:** 2024-05-28

**Authors:** Burcak Yavuz, Esra Cansever Mutlu, Zubair Ahmed, Besim Ben-Nissan, Artemis Stamboulis

**Affiliations:** 1Vocational School of Health Services, Altinbas University, 34147 Istanbul, Turkey; burcakyavuzz@gmail.com; 2Biomaterials Research Group, School of Metallurgy and Materials, College of Engineering and Physical Science, University of Birmingham, Birmingham B15 2TT, UK; e.mutlu@bham.ac.uk; 3Neuroscience & Ophthalmology, Institute of Inflammation and Ageing, University of Birmingham, Edgbaston B15 2TT, UK; 4Translational Biomaterials and Medicine Group, School of Life Sciences, University of Technology Sydney, P.O. Box 123, Broadway, NSW 2007, Australia; besim77@gmail.com

**Keywords:** nerve regeneration, non-coding RNAs, exosome, traumatic brain injury, blood-brain barrier

## Abstract

Extracellular vesicles (EVs), including exosomes, microvesicles, and other lipid vesicles derived from cells, play a pivotal role in intercellular communication by transferring information between cells. EVs secreted by progenitor and stem cells have been associated with the therapeutic effects observed in cell-based therapies, and they also contribute to tissue regeneration following injury, such as in orthopaedic surgery cases. This review explores the involvement of EVs in nerve regeneration, their potential as drug carriers, and their significance in stem cell research and cell-free therapies. It underscores the importance of bioengineers comprehending and manipulating EV activity to optimize the efficacy of tissue engineering and regenerative therapies.

## 1. Introduction

Extracellular vesicles (EVs) are defined as heterogeneous vesicles known to be evolutionarily conserved, originating from endosomes or the plasma membrane and released by cells [1,2]. They play active roles in both normal physiology and pathophysiology, owing to the significance of intercellular communication in prokaryotes and eukaryotic cells, which has driven their evolution [3]. EVs emerged as structures that facilitate this communication [4]. Today, it is known that EVs are encoded and released by the cell from which they originate and carry the characteristics of the cell. EVs were first thought to be procoagulant platelet-derived particles in plasma in 1946. Later, as a result of the study carried out in 1967 by Wolf, these structures began to be called platelet powder [5,6]. He thought these structures carried just cell residues that provided coagulation activity in those years. However, it was later realized that they had much more functional duties than carrying the cell debris. EVs consist of lipids, nucleic acids, proteins such as transmembrane and cytosolic proteins, and proteins involved with lipid metabolism. They are defined as lipid-bound vesicles excreted by cells into the extracellular space [7,8,9,10,11,12]. In general, a simple classification has been made according to release mechanisms and dimensions, but this classification is still not fully clarified. The studies to be carried out in the coming years will allow this classification to be updated.

Generally, a simple classification has been made according to release mechanisms and size. Studies to be carried out in the coming years will allow this classification to be updated. According to the minimal information for studies of extracellular vesicles (MISEV2023) guide published by the International Society for Extracellular Vesicles (ISEV) in 2024, the use of EVs subtypes continues to be carefully encouraged if they are differentiated according to properties such as size, density, molecular composition [13]. As studies continue for EV subclasses, new information is being obtained. EVs are recognised as a heterogeneous population of vesicles that vary in size, biological function, and origin. Published in 2024 and available on the Vesiclepedia website (http://www.microvesicles.org, accessed on 27 March 2024), the recognized types of EVs are roughly divided into small EVs and large EVs. An additional subtype, extracellular particles other than EVs (EPs), is also included in the classification of EVs. The classical classification includes exosomes (<200 nm), microvesicles (<1000 nm) and apoptotic vesicles (<5000 nm). Increasing studies on EVs have paved the way for discovering their new subtypes. The newly added subtypes, thanks to those studies are mitovesicles (<1000 nm), migrasomes (3000 nm), and exophers (<4000 nm) oncosomes (<1000 nm), megavesicles-large oncosomes (10,000 nm) [13,14,15,16,17,18,19,20,21]. EVs subtypes are shown in Figure 1 and Figure 2 below.

The newly discovered migrasomes, which are included in the classification of EVs subtypes, are thought to play a role in cell-to-cell communication through the release of enriched chemokines, morphogens or growth factors as carriers of damaged mitochondria or through lateral transfer of mRNA and protein [19,22]. The researchers found that migrasomes have a pomegranate-like structure and are rich in Tspan4 from the tetraspanin (Tspan) family in terms of protein content and that this protein may play an active role in its biogenesis [22,23]. In limited studies on migrasomes, it has been observed that they promote angiogenesis [24], provide mitochondrial homeostasis [25], and play a role in organogenesis and in the transfer of spatiotemporal chemical information required for left-right modelling [26]. It has also been observed that it mediates mitochondrial control and provides neutrophil viability [27]. With this physiological feature, we anticipate that it may be essential to combine future studies on migrasomes with vaccine technology. However, there is a need to elucidate its biogenesis and the mechanisms in which it plays a role.

Apoptotic bodies are released from dying cells and deliver useful material from the cell of origin to neighbouring healthy cells. Until this discovery, apoptotic bodies were thought to be not useful. Apoptotic bodies are formed when the membrane of an apoptotic cell bends outwards. It has been observed that their functional properties are the transfer of DNA fragments to phagocytes, cell survival, and inhibition of the inflammatory process [28,29]. Their size varies between 1 to 5 microns [30,31].

Like the migrasome, the newly discovered exophers are recognized as a new subtype of EVs with a size smaller than 4 microns [19]. The exosphere, which was thought to have an exosome-like release mechanism, was considered to be a different subtype of EVs due to its very large size compared to the exosome. Recent studies suggest that exophers actively eliminate neurotoxic aggregates when protein homeostasis is suppressed by proteotoxicity [32,33]. If the mechanism of eliminating aggregates against neuronal toxicity can be fully elucidated, we anticipate that it can be used to treat nervous system diseases. In studies published in 2020 and 2022, Nicolás-Ávila et al. showed that exospheres support the normal function of cardiomyocytes by playing a role in cardiac mechanisms [34,35].

Another type of EVs with an atypical size of 1–10 µm, released from cancer cells, are large oncosomes. These EVs are thought to contribute to metastasis after large oncosomes were found to be released by cancer cells and found in greater amounts in the plasma of invasive cancer cells and metastatic cancer [36].

In addition to these EV subtypes, in 2021, D’Acunzo et al. identified a distinct EV population following high-resolution density gradient separation of EVs isolated from Down syndrome and diploid control brains. This has led to identifying a third pool of metabolically competent mitochondrial-derived EVs in the EVs subtype. Studies using the ultracentrifugation method to isolate EVs have shown the presence of mitochondrial material in EV pellets centrifuged at 100,000× *g*. These vesicles have so far been interpreted as exosomes carrying mitochondrial proteins, lipids and mitochondrial DNA (mtDNA). Although it cannot be ruled out that some mitochondrial components may be present in exosomes, it has been reported that the vesicle was larger than 1 micron in findings with high mitochondrial evidence [37,38]. Like other novel vesicles, this vesicle is a novel EV subtype, the biogenesis and physiological processes of which need to be elucidated.

Among these EVs, the most studied sub-types are exosomes and microvesicles [39]. Exosomes and microvesicles have different properties according to their cargo contents Although exosomes and microvesicles are similar in size, their release pathways are entirely different. Exosomes are produced by the plasma membrane fusion of multivesicular endosomes (MVEs) followed by the release of intraluminal vesicles (ILVs), while microvesicles are secreted by outward vesiculation of the plasma membrane [40]. While various pathways exist for EV generation, the exact mechanism is still largely unknown. Cells can communicate with neighbouring cells or distant cells by the secretion of EVs and sub-types. All EVs have surface molecules that can target recipient cells [41]. After the EV binds to the target cell via a receptor-ligand interaction, intercellular signaling is induced, or it can enter the cell through endocytosis or phagocytosis. Thus, it triggers many changes to the physiological state of the target cell [42].

Exosomes and microvesicles have received considerable attention over the past decade (Figure 3 and Figure 4). It is, therefore, essential to elucidate their mechanisms and clarify the physiological processes in which they play an important role. This review will discuss the mechanisms of exosomes and microvesicles isolated from stem cells in the nervous system.

### 1.1. Exosome Biogenesis

Exosomes are bioactive vesicles secreted by cells, ranging in size from 40 to 200 nm, and taking an active role in intercellular cargo and communication, constituting the important subgroup of EVs. Exosomes are formed due to endosome differentiation [39,40]. Their nanoscale components play a sophisticated role in mRNA regulation and are associated with many physiological and pathological functions [41]. Subsequent studies have shown that exosomes function as a cell-to-cell communication mediator. They can easily interact with neighbouring cells to facilitate the transfer of active molecules [42,43]. Exosomes are made up of specially ordered proteins, lipids, and nucleic acids, depending on their cell type. It has been observed that approximately 4400 different proteins, 194 different lipids, 1639 different mRNAs, and 764 different miRNAs can be found in isolated exosomes of different origins to date. This shows that exosomes may have different contents from the cells they originate from. Specific components from different cell types can also be identified. These different components reveal the exosome’s complex structure and potential functional diversity [43,44].

The biogenesis of exosomes is highly complex [45]. Exosomes are generated from late endosomes formed by inward budding of the multivesicular body (MVB) membrane. Invagination of endosomal membranes leads to the formation of intraluminar vesicles (ILVs) within large MVBs. During the process, proteins are incorporated into the invaginating membrane, whereas cytosolic components are enclosed within ILVs. Upon fusion with the plasma membrane, the ILVs are released into the extracellular space, referred to as exosomes [46,47,48]. Exosome biogenesis can be summarized into steps I–IV: I—Cargo parsing into MVBs, II—MVB formation, III—Transport of MVBs, IV—MVB-plasma membrane fusion [49].

Currently, two kinds of pathways play a vital role in exosome biogenesis. The first is a dependent endosomal sequence complex transport (ESCRT) pathway, and the second is an independent ESCRT pathway [50,51]. These pathways may work synergistically, or different exosome subpopulations may be dependent on different mechanisms [52,53].

The ESCRT-dependent mechanism is one of the best-characterised mechanisms for the biogenesis of exosomes [51]. ESCRT consists of stepwise co-operating ESCRT-0, -I, -II, -III subcomplexes and proteins such as ATPase, VTA1, and Alix. ESCRTs are composed of class E vacuolar protein sorting (VPS) proteins assembled in complexes. ESCRT 0 consists of VPS 27/Hrs (yeast/human orthologues) and Hse1/STAM, while ESCRT-I consists of VPS 23/TSG101, VPS 28, VPS 37, Mvb12. ESCRT-II consists of VPS 22/EAP30, VPS 25/EAP25, VPS 36/EAP45, while ESCRT-III consists of VPS 20/CHMP6, SNF7/CHMP4, VPS 24/CHMP2 and VPS 2/CHMP3 [54,55,56,57]. ESCRT-0 interacts with ubiquitin to facilitate cargo aggregation and recruitment of other ESCRT complexes (ESCRT-I, ESCRT-II, and ESCRT-III) to the endosomal membrane. ESCRT-I and ESCRT-II are responsible for the budding of the endosomal membrane. Following ATP hydrolysis by VPS4, ESCRT-III subunits undergo sequential polymerization, ultimately driving membrane deformation and fission to produce ILV [58,59]. The Alix-dependent mechanism in the ESCRT-dependent pathway is the Syndecan-Syntenin-Alix pathway. The Alix task is to recruit ESCRT-III and VPS4 to complete ILV formation. In addition, Alix binds to CD9, CD63, and CD81 tetraspanins and can classify these proteins as biomarkers of ILVs/exosomes. Studies have also proven that heparan sulfate proteoglycans support exosome biogenesis through syntenin, a cytosolic adaptive protein [60,61,62].

In addition to the ESCRT-dependent pathway, which is the most known mechanism, an ESCRT-independent pathway has been discovered. This led to the realization that exosomes have a more complex structure than previously thought. The most known part of this pathway is the Neutral sphingomyelinase 2 (nSMase2)—ceramide pathway. This suggests that ceramides have a major role in membrane deformation in exosome release. [63]. IVL formation is regulated by sphingolipids, cholesterol, and proteins called lipid raft microdomains, while recently activated Rab31 GTPase has been shown to trigger membrane budding within these microdomains [64]. It has been suggested that these pathways involved in the biogenesis of exosomes may work synergistically (Figure 5) [65].

### 1.2. Ectosomes/Microvesicle Biogenesis

Microvesicles, which belong to the class of EVs, are larger compared to exosomes and represent a less studied subgroup of EVs. They typically range in size from 100 to 1000 nm [66,67].

The biogenesis of microvesicles differs significantly from that of exosomes. While exosomes are formed through the inward budding of the endosome, microvesicles can be released by outward budding from the plasma membrane [17].

Regarding the biogenesis of microvesicles, cytoskeletal reorganization and associated molecules play a crucial role. One such molecule is Ca^2+^. A substantial increase in Ca^2+^ levels triggers the activation of calpain, which breaks down cytoskeletal proteins. This, along with the modulation of flippase, floppase, and scramblase, allows for the remodelling of membrane asymmetry, facilitating the outward budding of microvesicles [66,68]. In coordination with myosin, actin induces a myosin-based cytoskeleton contraction following the cytoskeletal changes [69]. Through an ATP-dependent contraction mechanism, the coordinated action of actin and myosin leads to the separation of microvesicles from the plasma membrane [70,71]. Although the biogenesis of microvesicles involves more intricate processes, information on the intrinsic signals driving their formation remains limited. Microvesicles, also known as ectosomes, carry a diverse cargo, including cytosolic proteins, nucleic acids, metabolites, and plasma membrane proteins [72].

## 2. Stem Cell-Based EVs

Since Till and McCulloch pioneered the discovery of stem cells in 1961, their application in both scientific and clinical contexts has gained significant momentum in recent years, attracting growing interest [73,74]. Stem cells are unspecialized cells that can self-renew through division and differentiate into various cell types. They are crucial for basic and clinical research as they can proliferate to yield enough cells and differentiate into desired cell types based on their genetic potential. Moreover, they can integrate into the recipient’s tissue after transplantation [75,76]. Many adult stem cells undergo asymmetric cell division (ACD) to produce two daughter cells. In asymmetric cell division, one cell retains its stem cell identity, and another cell undergoes a differentiation program. This is particularly necessary for tissue homeostasis and regeneration [77]. ACD gives rise to another stem cell through self-renewal and a second cell type, which can be a differentiating progenitor or post-mitotic cell. In contrast to symmetric division, ACD allows stem cells to generate cellular diversity while maintaining a constant number of stem cells; thus, it can prevent accidental depletion or overgrowth of the stem cell population. This mechanism makes stem cells one of the most important elements, distinguished from other cells [78].

According to the stem cell hypothesis, the microenvironment plays a pivotal role in maintaining and directing stem cells. The preservation, differentiation, and repair of stem cells are intricately linked to the presence of this microenvironment, known as the stem cell niche [79]. The stem cell niche consists of stem cells, supportive cells, and scaffolds, collectively contributing to regulating the stem cell population and processes of determination, differentiation, and repair [80]. Advancements in stem cell technology have opened exciting prospects in regenerative and translational medicine. Stem cells can be broadly categorized according to origin into two classes: embryonic stem cells (ESCs) and mesenchymal stromal/stem cells (MSCs) [79,81,82]. ESCs, being pluripotent, have the potential to differentiate into cells from any of the three embryonic germ layers [83,84]. However, the use of embryonic stem cells is associated with certain challenges, including ethical concerns and the risk of teratoma formation. As an alternative, the utilization of programmable induced pluripotent stem cells shows great promise [85,86]. In contrast to embryonic stem cells, MSCs have more advantages despite their limited differentiation capacity [79,87,88].

There are several reasons why the use of MSCs is advantageous compared to embryonic cells:

Availability: MSCs can be obtained from various sources, such as bone marrow, adipose tissue, and umbilical cord blood, making them more readily accessible than other stem cell types [89];

Reduced immunogenicity: MSCs exhibit low immunogenicity, meaning they are less likely to elicit an immune response when transplanted into a recipient. This characteristic makes MSCs a suitable choice for allogeneic transplantation, where cells from a donor are used for therapy [90];

Immunomodulatory properties: MSCs possess immunomodulatory capabilities, allowing them to regulate immune responses and reduce inflammation. This feature is particularly beneficial in conditions where excessive immune responses contribute to tissue damage [91];

Tissue regeneration and repair: MSCs can promote tissue regeneration and repair through various mechanisms, including the secretion of growth factors, cytokines, and EVs. These factors stimulate endogenous repair processes and modulate the surrounding microenvironment [92];

Safety profile: MSCs have shown a favourable safety profile in preclinical and clinical studies, with a low incidence of adverse effects. This makes them a promising therapeutic option for many diseases and conditions [93];

Non-tumorigenic nature: MSCs have a lower risk of forming teratomas or other tumours, unlike embryonic stem cells. This characteristic enhances their safety profile and reduces concerns associated with tumorigenicity [93].

MSCs, however, also have disadvantages such as donor site morbidity, variability of reproduction and differentiation capacity, inadequate retention and differentiation of cultured cells, declining intrinsic activity and functionality of obtained cells over the donor’s age and inconsistent quality control for large-scale cell production [94,95,96,97,98].

These disadvantages are difficult to overcome, and focus has been placed on alternative cell-free therapies, specifically on using MSC stem cell-derived EVs. It is well-known that stem cells actively employ EVs in cell communication within their microenvironment. [94].

In addition, intercellular interactions mediated by MSC-derived EVs have been shown to play an important role in disease treatment. The bioactive molecules carried by EVs exert their effects on target cells through several mechanisms, including the direct stimulation of target cells via surface-bound ligands, transfer of activated receptors to recipient cells and EVs’ epigenetic reprogramming of recipient cells through the delivery of functional proteins, lipids, and non-coding RNAs [99]. Exosomes can engage in communication with both nearby and distant cells. Numerous studies have investigated the therapeutic efficacy of stem cell-derived EVs in various disease models.

## 3. Stem Cell-Based EVs and Nerve Regeneration

Regrowth of nerve fibres after injury or disease is essential for recovering lost function. The peripheral nervous system (PNS) can regenerate as it retains the intrinsic capacity to regenerate its axons, but the recovery is still not 100%, especially for large and complex PNS injuries. However, nerve regeneration in the central nervous system (CNS) of mammals fails for reasons which include a low intrinsic capacity to regenerate, apoptosis, scarring, lack of neurotrophic factors to support neurons and promote their axon regeneration and the presence of scar- and myelin-derived inhibitory molecules. CNS injuries, including spinal cord injury (SCI), traumatic brain injury (TBI) and optic nerve injury (ONI), therefore, represent a complex problem in biology with an unmet clinical need. Since this review focuses on the utility of EVs in nerve regeneration, the reader is directed to some excellent recent reviews on CNS and PNS regeneration elsewhere [100,101,102,103].

EVs have emerged as a possible alternative therapy to promote nerve regeneration. For example, EVs can transport much-needed signaling proteins and coding and non-coding RNA and thus facilitate multilevel communication. The most commonly used EVs in nerve regeneration have been derived from MSCs or other stem cells [104]. Hence, this review focuses on MSC-derived EVs to promote nerve regeneration. We searched PubMed, EMBASE and Web of Science for articles with the search string: stem cells AND extracellular vesicles OR exosomes AND nerve regeneration AND x, where x = peripheral nerve regeneration, traumatic brain injury, spinal cord injury, or optic nerve injury. There were no restrictions on publication date or language. All articles were screened by title and abstract for suitability. In total, and after duplicates were removed from the databases, 219 articles were eligible for full-text reading, with 144 articles dealing with PNS regeneration, 16 dealing with TBI, 42 dealing with SCI, and 17 dealing with optic nerve regeneration. Below is a summary of the key themes in each of these areas.

### 3.1. Stem Cell-Based EVs in Peripheral Nerve Injury (PNI)

Today, functional impairment and long-term disability resulting from nerve damage are among the most important clinical and public health problems. According to the literature, nerve damage is divided into three main groups: neuropraxia without axonal degeneration with localised myelin degeneration, axonotmesis with complete axonal interruption resulting in Wallerian degeneration, and neurotmesis with complete nerve interruption [105,106]. Although microsurgery is the preferred method for repairing these injuries, peripheral nerve repair often does not provide satisfactory functional recovery [107]. Therefore, alternative treatment methods are sought. Due to the contribution of stem cells to regeneration, especially in nervous system diseases and limited neuronal regeneration, studies on the use of stem cells in eliminating these damages have increased day by day. The therapeutic potential of stem cells and their underlying mechanisms largely depend on the conduction pathways and timing of transplantation, particularly in the context of peripheral nerve injuries, one of the most effective methods [108]. Numerous studies have demonstrated that systemic administration of stem cells in experimental stroke models reduces post-stroke brain damage, enhances neurological recovery, and activates neurodegenerative processes [109]. Recent studies have shown that EVs derived from stem cells instead of stem cells are highly effective in neuronal damage. These EVs have many advantages compared to stem cells, such as no risk of teratoma formation, low immunogenicity, ease of application, and the ability to innervate different parts of the nervous system from the site of infection [110]. MCS-derived EVs have been reported to regulate several genes to contribute to regeneration by reducing autophagy, showing anti-apoptotic, anti-inflammatory effects on damaged Schwann cell function, especially in peripheral injuries [111,112,113,114,115,116].

A study by Doeppner et al. showed that MSC-derived EVs ameliorated neurological disorders after ischemia similarly to MSCs, resulting in long-term neuroprotection associated with enhanced angiogenesis. On the other hand, MSC-EVs have been shown to act as immunomodulators against the post-ischemic suppressed immune system, contributing to creating a favourable external environment for nerve re-modelling [117]. In another study conducted in 2024, Bone Marrow Mesenchymal stem cell (BMSC) derived exosomes were obtained under hypoxic conditions, and proliferation in Schwan cells was examined in facial nerve injury among PNI. The results showed that BMSC-derived exosomes promoted proliferation and supported the paracrine function of Schwan cells through the circRNANkd2/miR-214-3p/mediator complex subunit 19 (MED19) axis. Researchers have also reported that this mechanism is promising for increasing its therapeutic potential in treating facial nerve [118].

Currently, acellular nerve allografts are known to support nerve regeneration. In exosome therapy isolated from hair follicle epidermal neural crest stem cells combined with acellular nerve allografts to close facial nerve defects, it has been shown that each rat has an improvement in behavioural changes and electrophysiological tests and has the potential to replace autograft therapy in the clinic by facilitating the healing of this damage to the facial nerve [119]. Moreover, decellularising nerve allografts with adipose-derived mesenchymal stem cells (ADSC) and exosomes isolated from these cells has been suggested to accelerate neuronal regeneration [120]. Although using ADSC-derived exosomes as a therapeutic target for peripheral nerve injury is promising, the mechanism is still poorly understood. However, miRNAs are known to play a role in Schwann cells’ proliferation and myelin regeneration. By knocking out ADSC-derived exosomes overexpressing miR-22-3p, we found that its effects on chromosome 10 (PTEN) activated the PI3K/Akt signaling pathway and contributed to the repair of the recurrent laryngeal nerve as well as the proliferation of Schwann cells [121,122].

In another recent study, to increase the therapeutic efficacy of MSCs, exosomes isolated from umbilical cord MSCs were cultured with exosomes derived from platelet-rich plasma containing abundant bioactive molecules (PRP-Exos), and these were found to play an active role in sciatic nerve injuries (SNI), especially in axonal regeneration and re-myelination. Researchers proved that in this case, where SNI activates the PI3K/Akt signaling pathway in the repair mechanism, PRP-Exos-MSCs promote neural repair and regeneration in PNI patients [123]. Exosomes isolated from dental pulp stem cells (DPSC-Exos) promoted axon and myelin regeneration after PNI by enhancing the migration of Schwann cells and secretion of neurotrophic factors in SNI [124].

Ma et al. reported that umbilical cord MSC-derived EVs effectively promote functional recovery and nerve regeneration in a rat sciatic nerve injury model, suggesting a promising clinical approach for peripheral nerve repair [125]. Indeed, researchers have suggested that the clinical application of MSC-derived EVs may be more appropriate than the application of stem cells. In an in vivo study by Shiue et al., Human Umbilical Cord MSC-derived exosomes were injected intrathecally for pain caused by nerve damage following spinal cord nerve ligation. It was reported by the researchers that the thermal and mechanical sensitivity caused by pain decreased and that it contributed to neuronal development by increasing glial cell-derived neurotrophic factor (GDNF), brain-derived neurotrophic factor (BDNF), and IL-levels [126]. In another in vivo study, exosomes isolated from adipose-derived stem cells increased regeneration in peripheral nerve crush [127]. Liu et al. injected bone marrow stem cell-derived exosomes in a traumatic spinal cord injury. At the end of the study, it was shown that MSC-derived exosomes supplied in various doses could suppress glial scars, reduce neuronal cell apoptosis, reduce the inflammatory response, promote axonal regeneration, and consequently support post-acute functional behavioural recovery [128]. Limb motor dysfunction, i.e., partial paralysis or paralysis, which is seen as the most serious problem of peripheral injuries, is still considered a more important risk regarding patients’ quality of life and health [129]. Webb et al. reported that neural tissue preservation and functional levels improved in the EVs hemiplegia disease model obtained from neuronal stem cells [130]. Liu et al. confirmed these results last year [131].

Similar studies pull out the role of stem cell-derived EVs, which can facilitate the creation of a suitable micro-environment for nerve regeneration by activating the secretion of neurotrophic and nerve growth factors, thereby promoting the regeneration of the damaged neuron [132,133,134]. Therapeutic mechanisms in MSCs-EVs applications in peripheral injuries include sciatic nerve, neuropathy due to different reasons, and partial or complete paralysis. It has been observed that molecular pathways such as modulating neuroinflammation [135], regulating the TSG-6/NF-κB/NLRP3 signaling pathway involved in immunomodulatory function [136], regulation by Schwann cell c-Jun [137], and restoration of the level of Neurotrophin-3, a brain-derived neurotrophic factor, are used [138].

Today, clinical studies of stem cell-derived EVs are expanding, and their role in neural regeneration is considered a promising tool and a less expensive therapy [139]. The therapeutic efficacy of MSCs-EVs in both in vivo and in vitro models of nerve injury has been proven by several studies. However, applying this technology in clinical trials remains challenging, and no reliable reports have been presented.

### 3.2. Stem Cell-Based EVs in Traumatic Brain Injury (TBI)

TBI affects over 60 million people worldwide and can be caused by various factors, such as motor vehicle accidents, falls and sports-related injuries. TBI results in physical and cognitive impairments such as paralysis, loss of sensation, coordination, memory loss, difficulty concentrating and impaired judgement [140,141]. Since CNS axons fail to regenerate after brain injury, TBI results in significant economic burdens, including medical expenses, lost productivity and long-term care needs [142]. TBI results in neuroinflammation, glial scar formation and neurodegeneration from increased pro-inflammatory cytokines, astrocyte activation and damage to the blood–brain barrier (BBB) [143,144,145]. Since there are no treatments for TBI that promote the regeneration of axons in the brain, various strategies have been proposed to help neurons regenerate their axons. Stem cell-derived exosomes have been suggested as a possible therapy to promote regeneration in the injured brain. However, despite their promise, none of the therapies have been used for clinical trials in TBI, so only preclinical studies in animal models are discussed.

In analysing the studies on TBI, it appears that the main beneficial effects of exosomes in TBI are attributable to their paracrine effects. Several studies have shown that MSC-derived exosomes can improve neurobehavioural performance, promote neurogenesis and angiogenesis and reduce inflammation after TBI. Exosomes for the treatment of TBI have been derived from a variety of sources, including MSCs, bone marrow-derived stem cells (BMSCs), olfactory ensheathing cells (OECs), human umbilical cord MSCs (HUCMSCs), neural stem cells (NSCs) and pericytes. For example, after a controlled cortical impact (CCI) to the exposed brain hemisphere followed by retro-orbital injection of BMSC-derived exosomes, neurobehavioral outcomes assessed using the modified neurologic severity score (mNSS) were significantly improved in exosome-treated groups compared to PBS treated controls at both 7 and 14 days after TBI [146]. Likewise, animals also showed improvements in performance on the rotarod test at 14 days post TBI, reduced lesion area, reduced apoptosis, reduced inflammation and reduced activation of microglia/macrophages of the M1 phenotype, whilst increasing M2 pheno-types after TBI [147]. M2 macrophages are thought to be beneficial in the CNS and promote axon regeneration by boosting the intrinsic capacity of CNS neurons to regenerate, thus promoting functional recovery after TBI [148].

Mesenchymal stromal cell-derived exosomes in a similar CCI injury model significantly improved spatial learning, analysed by the Morris Water Maze test and neurological function, assessed by mNSS and forelimb footfault test, when administered or systemically [149,150]. In addition, mesenchymal stromal cell-derived exosomes significantly increased vascular density and angiogenesis, neurogenesis in the dentate gyrus, and reduced brain inflammation after TBI [149]. Human MSC-derived exosomes also exert benefits after TBI by suppressing neuroinflammation and thus improved pattern separation function (the ability to discriminate analogous experiences through storage of similar representation in a non-overlapping manner—a function that is reliant on hippocampal neurogenesis) and spatial learning ability [151].

Another developing area of exosome research is their ability to secrete microRNAs, which are a class of small non-coding RNA molecules that regulate a variety of gene expression at the post-transcriptional level by binding to complementary seed sequences of target mRNAs, leading to degradation or translational repression [152]. Several studies have shown the importance of miRNAs in regulating cellular processes such as neuronal apoptosis, angiogenesis, the immune-inflammatory response and BBB permeability after TBI [153,154,155,156,157]. For example, expression of miR-21 mRNA levels in a TBI group peaked at 3 days after injury compared to sham-treated controls, whilst upregulation of miR-21 in the brain significantly improved neurological outcomes (in both mNSS and Morris Water Maze tests) [154]. Likewise, increased miR-124-3p in microglial-derived exosomes after a repeat TBI (rTBI), increased neurite outgrowth and improved neurological outcomes in rTBI mice [158]. The positive effects of miR-124-3p are thought to be due to protective effects on injured neurons through inhibition of autophagy [159]. Other miRs such as miR-21-5p, miR-216a-5p, and miR-210 regulate neuroprotection, improve mitochondrial function and inhibit lipid peroxidation in vascular endothelial cells after TBI [160,161,162].

In summary, most of the effects of stem cells in TBI arise from their secretome, including exosomes. The promising results on neuroprotection and functional recovery remain to be validated in human trials.

### 3.3. Stem Cell-Based EVs in Spinal Cord Injury (SCI)

SCI affects 1.2 million people worldwide annually and leads to loss of sensation motor and autonomic function [163]. Patients with SCI present with comorbidities such as bladder, sexual, gastrointestinal and respiratory dysfunction, as well as recurrent urinary tract infections, in severe cases, can result in death [164]. Like TBI, the leading causes of SCI include motor vehicle accidents, falls and sporting injuries, leading to long-term physical disability and psychological problems, requiring lifelong care [165,166,167]. The permanent disability after SCI is due to the inability of the axons to regenerate and reconstruct lost neural circuits [168]. The cellular microenvironment of the SCI site, which consists of nerve cells, astrocytes, microglia, and oligodendrocytes, undergoes rapid changes in both phenotype and function. The primary injury causes neurons to die by necrosis and apoptosis, astrocytes to undergo overactivation due to inflammation, microglia to become activated and cause BBB breakdown, whilst severed axons undergo Wallerian degeneration, leading to loss of function [169,170,171,172,173,174]. Mature NG2+ oligodendrocytes are not only reduced in number but display altered morphology at the lesion site that results in reduced myelination, whilst oligodendrocyte precursor cells (OPCs) are activated and secrete axon growth inhibitory chondroitin sulphate proteoglycans (CSPGs) [175]. As well as this, the degenerating myelin sheath unravels myelin-derived axon growth inhibitory molecules such as Nogo, oligodendrocyte-derived myelin glycoprotein (OMgp), myelin-associated glycoprotein (MAG) [102].

To date, no treatments can regenerate axons completely after SCI. However, some progress has been made in understanding the reasons for the failure of CNS axons to regenerate, and this has been translated into preclinical models of SCI that show some promise [176,177]. For example, research in inflammation, scar formation, cell transplantation, and biomaterials to promote repair have all demonstrated some success in preclinical models of SCI. More recently, combinations of biomaterials, stem cells, growth factors and drugs have led to improvements in axon regeneration over single treatments [178]. For more information on these research areas, the reader is directed to some excellent reviews elsewhere [102,179,180,181,182,183].

EVs in the CNS may be novel “non-cellular” therapy and are known for their ability to exchange information and play a part in functional activities in the CNS [184]. MSC-derived EVs (MSC-EVs) are rapidly gaining traction in promoting SCI repair, both independently and as combinatorial treatments [185]. In general, EV research in SCI has focused on three main areas: (1) inhibition of inflammation; (2) activation of axon regeneration or reconstruction of damaged circuits to promote functional recovery; and (3) their combination with biomaterials to support growth factor delivery and guided regeneration [186,187].

The first step to regenerating the axons of the spinal cord is to improve the micro-environment of the damaged nerve. One way to achieve this is to suppress acute inflammation, making the microenvironment more conducive to axon regeneration. For example, suppressing inflammation using MSC-EVs that target the activation of nuclear factor-κB (NF-κB) signaling and microglial complement system promoted axon regeneration and functional recovery after SCI [188]. Likewise, various miRNAs secreted by MSC-EVs, such as miR-21-5p, miR-182 and let-7b, which polarise macrophages towards the anti-inflammatory M2 phenotype, can alleviate inflammation mediated by toll-like signaling and promote axon regeneration and functional recovery [189,190,191]. The recruitment of macrophages from the peripheral circulation in response to vascular damage further exacerbates inflammation and is detrimental to axon regeneration, thus making it an important target after SCI [192]. MSC-EVs also protected pericytes, components of the neurovascular unit, and ultimately improved motor function recovery [193,194]. In addition, neuron-derived exosomes can change the phenotype of astrocytes and microglia in the injury site and modulate neighbouring neurons’ activity, thus promoting axon regeneration and functional recovery after SCI [192,195,196].

Activating the intrinsic neuronal growth programme normally repressed in differentiated neurons is vital in promoting axon regeneration after SCI [182]. The mammalian target of the rapamycin (mTOR) signaling pathway has emerged as an important regulator of neuronal growth [100]. The mTOR pathway regulates cell growth, metabolism and protein synthesis. In adult CNS neurons, mTOR is kept in check by increased activity of phosphatase and tensin homolog deleted on chromosome ten (PTEN), a negative regulator of mTOR, which results in attenuated axon regeneration [197]. However, activating the mTOR pathway results in extensive axon regeneration and functional recovery after SCI [198,199]. MSC-EVs can also inhibit PTEN because their cargo is composed of miR-21 and miR-19b, thus promoting axon regeneration and functional recovery after SCI [200,201]. MSC-EVs can also stimulate axon regeneration after SCI by promoting the proliferation of endogenous stem cells through the ERK pathway [202,203,204].

Combined MSC-EVs and biomaterials, such as hydrogels, are being used to improve SCI repair. This relies on support functions by hydrogels and their ability to slowly release growth factors over time, guiding axonal regeneration through the lesion site [205,206]. Encapsulating MSC-EVs in an injectable anti-inflammatory hydrogel not only prolonged the release of EVs but also improved motor functional recovery after SCI [207]. The conductive properties of neuronal tissues have been mimicked using a conductive hydrogel composed of gelatin methacrylate and polypyrrole-containing EVs, which induced differentiation of neural stem cells in vitro and enhanced growth of myelinated axons after SCI [208]. These studies demonstrate the beneficial effects of combining biomaterials with EV treatment to promote axon regeneration and functional recovery after SCI.

Despite the promise of MSC-EVs in SCI, there are many limitations to overcome before MSC-EVs can be used in large-scale applications. Stem cells lose their “stem cell” properties under conventional culture conditions and undergo senescence, leading to reductions in their number and therapeutic efficacy [209,210]. MSC-EVs are also poorly targeted to the SCI site after intravenous application, which is the preferred method of drug delivery after SCI, and hence, both the production of EVs and their targeting capacity need to be improved before clinical application [211,212,213]. Methods such as pretreating MSC-EVs, e.g., in hypoxic conditions, maintain their stem cell properties and enhance their ability to proliferate [214]. In addition, modification of MSC-EVs by engineering means such as piggybacking or membrane modification, may enhance the targeting of SCI sites [215,216].

### 3.4. Stem Cell-Based EVs in Optic Nerve Injury (ONI)

The optic nerve is an extension of the CNS, and like other areas of the CNS, injury or disease causes neuronal death, dysfunction and thus loss of function. The optic nerve transmits visual signals from the eye to the visual centres in the cerebral cortex. Damage to the optic nerve can occur via direct ocular trauma and diseases such as glaucoma, diabetic retinopathy, optic neuritis and ischemic optic nerve diseases, leading to optic atrophy, degeneration, vision loss and blindness [217,218,219]. ONI causes rapid loss of retinal ganglion cells (RGC) that reside in the retina, with their axons forming the optic nerve, and as such, have been the subject of many studies on neuroprotective therapies. These include anti-apoptotic factors such as anti-caspase treatments [220,221,222,223,224] and growth factors such as brain-derived neurotrophic factors [225,226]. However, EVs have emerged as a new treatment potential for ocular injury and disease.

Recent studies have highlighted that exosomes released by MSCs, particularly miRNAs, play a pivotal role in RGC neuroprotection and axon regeneration. For example, when inhibited, miRNA-linked protein Argonaut 2 (Ago2), which regulates the biological function of miRNAs, reduced RGC neuroprotection and axon regeneration [227]. Another study confirmed that exosomes isolated from bone marrow stem cells promoted RGC neuroprotection and prevented retinal nerve fibre layer thinning and loss of function, but when Ago2 was knocked down, all of these effects were significantly attenuated [228]. Micro RNAs can also impact axon growth signaling pathways to regulate RGC axon regeneration and apoptosis. For example, miR-93-5p targets PTEN and regulates NMDA-induced autophagy in RGC during glaucoma [229]. Down-regulation of miR-19a in optic nerve crush injury promotes RGC axon regeneration, effectively relieving PTEN inhibition and promoting axonal elongation [230]. miR-132 also participates in axonal elongation as it targets the Rho GTPase activating protein, p250GAP, which normally inhibits the formation of RGC branches [231].

Other miRNAs, such as miR-187, miR-708, miR-335-5p, miR-149 and miR-211, can all promote RGC neuroprotection after ONI and glaucoma and represent important molecules for further downstream development and translation into the clinic. Viral vector-mediated delivery of six candidate miRNAs (miR-26a, miR-17, miR-30c-2, miR-92a, miR-292, and miR-182), several of which target the PTEN pathway, was shown to enhance RGC neuroprotection and axon regeneration [232]. Likewise, exosome-mediated delivery of the neuropeptide, pituitary adenylate cyclase-activating polypeptide 38 (PACAP38) significantly enhanced RGC survival and axon regeneration [233]. Currently, most of the work on exosomes after ONI revolves around their miRNA cargo; however, the background specificity of exosomes is poorly understood. There are also differences in the composition of exosomes from different stem cell sources, and their effects are also variable. Hence, a greater understanding of EVs in ONI will improve their potential use as therapeutics in ocular injury and disease.

## 4. Challenges and Limitations of Using EVs for Therapeutic Purposes in Neuroregeneration

In vitro and in vivo studies of EVs have demonstrated their therapeutic efficacy and potential in nerve regeneration, especially in the CNS, which has limited nerve regeneration capacity and may be a source of hope for currently incurable nervous system diseases. Although SC-EVs use in nervous system injury and diseases is still quite limited, the interest in this field has increased. Despite some studies showing therapeutic efficacy, no approved and clinically applied treatment methods exist. We believe that the lack of an approved treatment is due to several problems that need to be overcome. Firstly, there are some disadvantages to the use of SC-derived EVs in treatment, including the following:SC-EVs have short half-lives;SC-EVs have limited targeting capacity;SC-EVs may be insufficient in content (DNA, miRNA, proteins, etc.);BBB transition not yet fully proven;Accumulation in the lungs, liver, and spleen has been reported [234,235].

In addition to these disadvantages, different problems arise depending on the source from which the EVs are obtained. In addition to these limitations, the use of EVs isolated in vitro from stem cells, especially for patients with CNS diseases, suffers from donor heterogeneity and is one of the most critical problems in the application of MSCs and EV therapies in the clinical setting [236]. Moreover, other problems must be overcome in the clinical use of EVs. These challenges include:

*1-Standardisation*: Establishing standardised protocols for the isolation and characterisation of EVs to ensure consistent quality and reproducibility of therapeutic preparations is crucial. Different populations and subpopulations of EVs need to be isolated according to their biophysical and biochemical properties. However, current separation methods are complex, and further studies are needed, especially regarding EVs isolated from cell culture media [237,238]. Developing a standard protocol for EVs to be obtained from stem cells is necessary to pave the way for their use in medicine. The efficiency of EVs obtained from sequestrants collected from different MSCs from cell culture also varies. Even the isolation method plays a critical role in EV efficiency. For example, various methods have been used for the isolation and purification of EVs, including:Differential Centrifugation/Ultracentrifugation with or without Sucrose Gradient Cushion;Polymeric Precipitation Isolation;Size Exclusion Chromatography;Immunoaffinity Isolation;Ultrafiltration And Microfiltration Technologies;Microfluidic Devices;EVs Isolation Reagents [14,15,239,240,241].

Among these methods, the most frequently investigated method is ultracentrifugation. This method allows isolation from large-scale samples, especially in cell culture. However, in recent years, there has been evidence that high-speed centrifugation (100,000× *g*) may affect EV structure and function [242]. Ultracentrifugation is also a costly method in terms of equipment, but sucrose density gradients can also be used in centrifugation-based methods to improve isolation. However, a disadvantage of this method is the need for additional purification to separate EV subpopulations from other microparticles with similar densities and the density gradient matrix [243]. Combined methods may provide higher yields while maintaining EVs’ biophysical and protective properties, often resulting in more successful isolation, especially from stem cell cultures [244]. Bringing together relevant standardisation studies and establishing standard protocols to ensure that the sample obtained in EV isolation has a minimum contamination rate is important in the clinical approval and implementation phase. Another important element for standardisation is the need to characterise isolated EVs and establish protocols at this stage. Although the characterisation of the sample obtained after isolation, concentration, and size analysis can be performed, detailed characterisation of the molecular composition of EVs and EV subpopulations is still a challenge [245]. Protocols used to measure size and concentration have primarily been established. Further molecular analysis protocols are also needed to examine the efficiency of EV isolation in terms of the degree of purity and the mechanisms that may affect the target tissue, biodistribution, and regeneration. Particular challenges for the characterisation of EVs are low specificity and sensitivity.

*2-Storage Optimisation*: Optimal storage conditions for SC-EVs must be determined to maintain their stability and therapeutic efficacy during storage and transportation. Evidence suggests that −80 °C is the optimal temperature to preserve EV content for downstream molecular profiling. This temperature has shown its usefulness in protecting EVs, as it is effective in long-term storage. However, refreezing of thawed EVs for use and the continuation of this cycle has been found to cause significant losses in the structure and content of EVs. Therefore, the importance of avoiding a few freeze–thaw cycles needs to be emphasised [246].

*3-Safety of SC donors and isolated EVs*: Here, it is also necessary to pay attention to the donors of MSCs at the source of their acquisition, especially in cases where cultured EV isolation is performed. To standardise the MSC recovery, the effect of variability in MSC donor characteristics during production on the isolation process should also be considered [247]. Recent studies have found that EVs carry the characteristics of cells from young or old donors and transmit them to target cells. In this case, epigenetic modification of genes and the composition of miRs associated with SC proliferation, self-renewal, and differentiation patterns can be observed. Changes in target cells are thought to be caused by either mRNA transcripts or miRs transferred by EVs. However, it has been emphasised that other cargoes (such as proteins and metabolites) within the EV may also be involved in this process [248]. For this reason, the trophic structure and cytokine profiles of MSCs may depend on the donor’s individuality and directly affect the therapeutic efficacy of the EVs isolated from them. Good Manufacturing Practice (GMP) conditions and standardisation are required to minimise donor changes and their effects.

In addition, SC-EVs need to be sterilised after the isolation phase and before application. In this regard, EVs are generally separated from possible pathogens by subjecting them to filtration. Ensuring the sterility of EVs without compromising their content and structural integrity is important for safe clinical applications.

*4-Scalable Production*: Many studies on SC-EV isolation have generally been performed using cell culture techniques. However, the biggest problem encountered in cell culture techniques, especially in two-dimensional cultures, is that long-term passage to produce sustainable amounts of EVs may cause cells to lose their clonal and differentiation capacity [249]. Although methods have been developed for SC culture techniques to increase efficiency and avoid existing problems, current EV production methods still have lower yields. This hinders the advancement of preclinical and clinical use of EV as a therapeutic [250]. Therefore, developing large-scale production techniques is necessary to meet the increasing demand for SC-EVs in clinical settings. In studies conducted to increase the efficiency of EVs, it has been pointed out that the efficiency is higher in a three-dimensional culture environment [251]. This demonstrates that maintaining the cells in the spheroid structure may release more bioactive molecules into the local environment, maintaining EV efficiency.

In the culture of SCs, several protocols increase the efficiency of EVs. For example, culture under GMP conditions, especially in a serum-free medium, maintains SC morphology, phenotype and viability while triggering an efficient EV release. Therefore, SCs offer large-scale production options when GMP conditions are met. In addition, manipulating EV-biogenesis biology allows for manipulating culture conditions and increased EV yields [252,253].

Although there are still challenges to overcome, the ability to design EVs with recent studies may allow these problems to be solved to some extent. Studies such as increasing the half-life of EVs, surface modifications of EVs for targeting [254,255], and increasing the content richness of EVs as carrier systems show that SC-derived EVs can be used actively in therapeutic terms. [241]. Considering SC-EVs’ efficacy in peripheral nerve injuries, their contribution to nerve regeneration in the CNS cannot be ignored [256].

## 5. Conclusions

There is growing interest in clinical trials investigating the therapeutic effect of MSCs in various target systems and their potential applications. While the ethical concerns and risk of teratoma associated with stem cells still impact clinical studies, using stem cell-derived EVs shows promise in overcoming these limitations. EVs possess similar therapeutic capabilities to stem cells but without many drawbacks associated with cell-based therapies. These unique properties position EVs as a valuable therapeutic approach. In the future, EVs may be utilized not only for therapeutic purposes but also for diagnostic applications. With their ability to regulate gene expression, facilitate tissue regeneration, and exert anti-inflammatory effects, EVs can potentially become highly effective therapeutic agents.

Moreover, by modifying EVs, their therapeutic efficacy can be further enhanced through targeted delivery using nanocarrier systems. Stem cell-derived EVs can spread therapeutic properties locally and systemically, making them a safer alternative to cellular therapy and transplantation surgery. As research in this field deepens in the coming years, stem cell EVs are expected to offer significant advancements in regenerative medicine.

Once these challenges are overcome, future research can focus on developing bioengineered MSCs with enhanced properties for targeted delivery of various therapeutic molecules and increased yield of EVs. We anticipate that there will be significant advancements in this field, with numerous new studies emerging in the coming years.

## Figures and Tables

**Figure 1 ijms-25-05863-f001:**
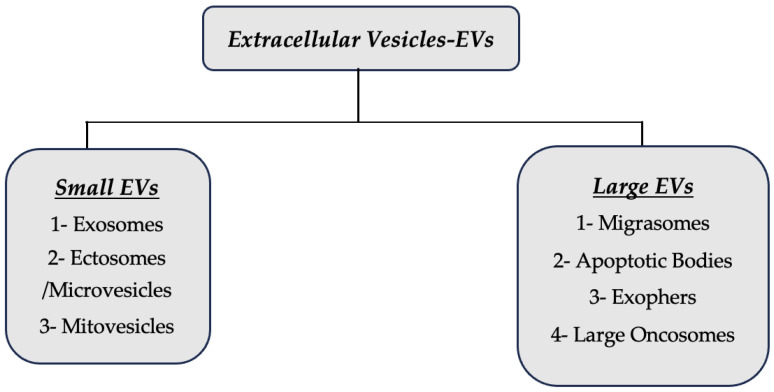
EVs subtypes.

**Figure 2 ijms-25-05863-f002:**
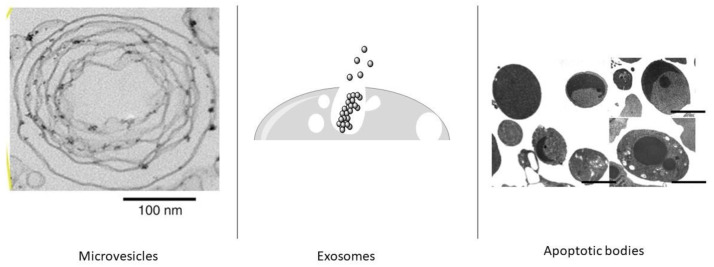
Schematic representation of microvesicles [12], exosomes [13] and apoptotic bodies [15].

**Figure 3 ijms-25-05863-f003:**
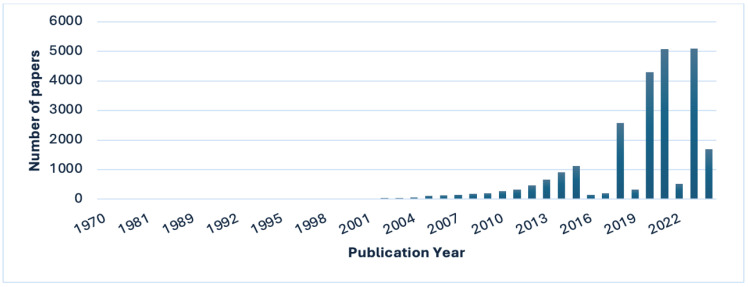
Bar chart of the number of publications per year after searching for “exosomes” in the PubMed search engine (search was performed on 27 March 2024).

**Figure 4 ijms-25-05863-f004:**
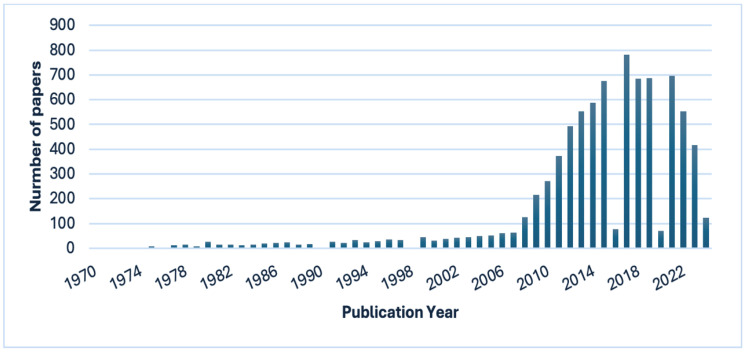
Bar chart of the number of publications per year after searching for “microvesicles” in the PubMed search engine (search was performed on 27 March 2024).

**Figure 5 ijms-25-05863-f005:**
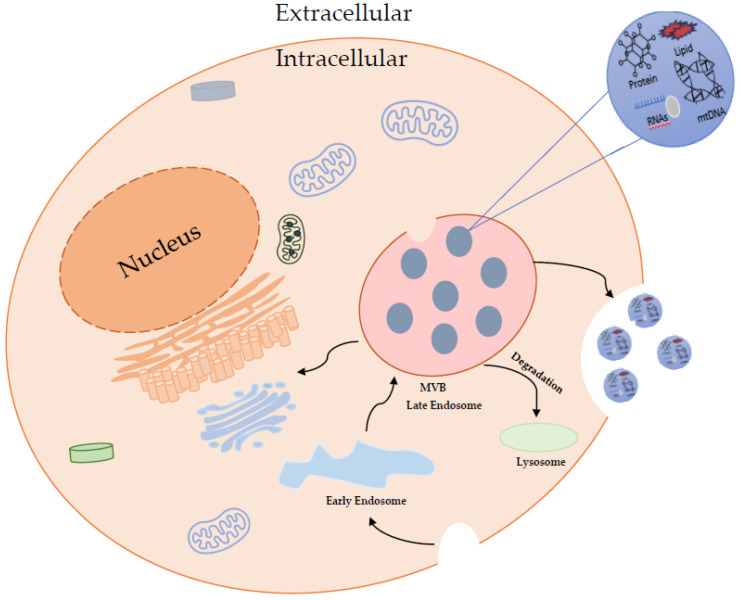
Exosome Biogenesis. The exosome development begins with endocytosis to form early endosomes, leading to late endosomes by inward budding and multivesicular bodies (MVBs). MVBs could then degrade and generate lysosomes or fuse with the cell membrane and release ILVs that become exosomes in the extracellular environment (Figure adapted from [65]).

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
