# Peer review of "Applications of Stem Cell-Derived Extracellular Vesicles in Nerve Regeneration"

_ijms, 2024, doi:10.3390/ijms25115863_

Round 1

Reviewer 1 Report (Previous Reviewer 3)

Comments and Suggestions for Authors

The manuscript, aimed at illustrating the applications of stem cell-derived extracellular vesicles in nerve regeneration, included much information, with an interesting attention to translational applications, including clinical trials. However, inaccuracies and unclear points should be addressed

Major points

Line 34 The sentence “In these conditions, EVs can be defined as the fingerprint of the cell in another sense.” is awkward.

Line 46: Authors wrote that “… it is seen that there are 6 different types of EVs. These are, exomeres [12]…”. Exomers are lipid nanoparticles but cannot be classified as vesicles, as they are not surrounded by a lipid bilayer.

Line 76-77 It is unclear from the reference if giant plasma membrane vesicles are natural or artificial and what is the basis of a different classification with respect to microvesicles and large oncosomes

Line 281 Authors specified that only “Some of these studies are summarized in Table 2.” It is acceptable that only some studies are summarized, but authors should clarify the inclusion criteria.

Line 349 the sentence “However, applying this technology in clinical trials remains challenging and no reliable reports have been presented.” Should be discussed, clearly specifying what is still challenging in this promising technology, or referring to Section 2.3

In section 2.3 points 1 and 3 appears overlapping. Please clarify

Author Response

Response to Reviewer 1

Thank you very much for the constructive comments you made to help us improve our review paper. Below, you can find our answers to your comments:  

  1. The manuscript, aimed at illustrating the applications of stem cell-derived extracellular vesicles in nerve regeneration, included much information, with an interesting attention to translational applications, including clinical trials. However, inaccuracies and unclear points should be addressed

Major points

  1. Line 34 The sentence "In these conditions, EVs can be defined as the fingerprint of the cell in another sense." is awkward.

We modified the sentence. Thank you.

  1. Line 46: Authors wrote that "... it is seen that there are 6 different types of EVs. These are, exomeres [12]...". Exomers are lipid nanoparticles but cannot be classified as vesicles, as they are not surrounded by a lipid bilayer.

The sentence is now deleted. We also added the detailed EV classification as seen according to MISEV guideline 2023 (see below).

In general, a simple classification has been made according to release mechanisms and size. Studies to be carried out in the coming years will allow this classification to be updated. According to the minimal information for studies of extracellular vesicles (MISEV2023) guide published by the International Society for Extracellular Vesicles (ISEV) in 2024, the use of EVs subtypes continues to be carefully encouraged if they are differentiated according to properties such as size, density, molecular composition [13]. As studies continue for EVs subclasses, new information is being obtained. EVs are recognised as a heterogenous population of vesicles that vary in their size, biological function, and origin. Published in 2024 and available on the Vesiclepedia website (http://www.microvesicles.org), the recognised types of EVs are roughly divided into small EVs, large EVs. An additional subtype, extracellular particles other than EVs (EPs), is also included in the classification. EVs. The classical classification includes exosomes (<200 nm), microvesicles (<1000 nm) and apoptotic vesicles (<5000 nm). Increasing studies on EVs have paved the way for the discovery of their new subtypes. The newly added subtypes, thanks to those studies, are microvesicles (< 1000 nm), microsomes (3000 nm), others (< 4000 nm), endosomes (<1000 nm), mega vesicles-large endosomes (10000 nm) [13- 21]. EVs subtypes are shown in Figures 1 and 2 below.

 The newly discovered migrasomes, which are included in the classification of EVs subtypes, are thought to play a role in cell-to-cell communication through the release of enriched chemokines, morphogens or growth factors, as carriers of damaged mitochondria or through lateral transfer of mRNA and protein [19, 22]. The researchers found that migrasomes have a pomegranate-like structure and are rich in Tspan4 from the tetra-spanin (Tspan) family in terms of protein content and that this protein may play an active role in its biogenesis [22, 23]. In limited studies on migrasomes, it has been observed that they promote angiogenesis [24], provide mitochondrial homeostasis [25], and play a role in organogenesis and in the transfer of spatio-temporal chemical information required for left-right modelling [26]. It has also been observed that it mediates mitochondrial control and provides neutrophil viability [27]. With this physiological feature, we anticipate that it may be important to combine future studies on migrasomes with vaccine technology. However, there is a need to elucidate its biogenesis and the mechanisms in which it plays a role. 

Apoptotic bodies are released from dying cells and deliver useful material from the cell of origin to neighbouring healthy cells. Until this discovery, apoptotic bodies were thought to be useless. Apoptotic bodies are formed when the membrane of an apoptotic cell bends outwards. Their functional properties include the transfer of DNA fragments to phagocytes, cell survival, and inhibition of the inflammatory process [28, 29]. Their size varies between 1 and 5 microns [30, 31].

Like the migrasome, the newly discovered exophers are recognised as a new subtype of EVs with a size smaller than 4 microns [19]. The exosphere, which was thought to have an exosome-like release mechanism, was considered to be a different subtype of EVs due to its very large size compared to the exosome. Recent studies suggest that exospheres play an active role in eliminating neurotoxic aggregates when protein homeostasis is suppressed by proteotoxicity [32, 33]. If the mechanism of eliminating aggregates against neuronal toxicity can be fully elucidated, we anticipate that it can be used in the treatment of nervous system diseases. In studies published in 2020 and 2022, Nicolás-Ávila et al. showed that exophers support the normal function of cardiomyocytes by playing a role in cardiac mechanisms [34, 35].

 (see Figure 2 in the document)

 Figure 2. Schematic representation of microvesicles [12], exosomes [13], and apoptotic bodies [15]. Another type of EVs with an atypical size of 1-10 µm, released from cancer cells, are large oncosomes. These EVs are thought to contribute to metastasis after large oncosomes were found to be released by cancer cells and found in greater amounts in the plasma of invasive cancer cells and metastatic cancer [36]. 

 In addition to these EVs subtypes, in 2021 D' Acunzo et al. identified a distinct EV population following high-resolution density gradient separation of EVs isolated from Down syndrome and diploid control brains. This has led to identifying a third pool of metabolically competent mitochondrial-derived EVs in the EVs subtype. Studies using the ultracentrifugation method to isolate EVs have shown the presence of mitochondrial material in EV pellets centrifuged at 100,000 x g. These vesicles have so far been interpreted as exosomes carrying mitochondrial proteins, lipids and mitochondrial DNA (mtDNA). Although it cannot be ruled out that some mitochondrial components may be present in exosomes, it has been reported that the vesicle was larger than 1 micron in findings with high mitochondrial evidence [37, 38]. Like other novel vesicles, this vesicle is a novel EVs subtype, the biogenesis and physiological processes of which need to be elucidated.

  1. Line 76-77 It is unclear from the reference if giant plasma membrane vesicles are natural or artificial and what is the basis of a different classification with respect to microvesicles and large oncosomes

We modified the text. See below:

Another type of EVs with an atypical size of 1-10 µm, released from cancer cells, are large oncosomes. These EVs are thought to contribute to metastasis after large oncosomes were found to be released by cancer cells and found in greater amounts in the plasma of invasive cancer cells and metastatic cancer.

  1. Line 281 Authors specified that only "Some of these studies are summarised in Table 2." It is acceptable that only some studies are summarised, but authors should clarify the inclusion criteria.

We deleted Table 2, but instead, we provided Figures 2 and 3, as seen in the document and added the following explanation:

Exosomes and microvesicles have received considerable attention over the past decade (Figures 3 and 4). It is, therefore important to elucidate their mechanisms and clarify the physiological processes in which they play an important role. This review will discuss the mechanisms of exosomes and microvesicles isolated from stem cells in the nervous system.

  1. Line 349 the sentence "However, applying this technology in clinical trials remains challenging and no reliable reports have been presented." Should be discussed, clearly specifying what is still challenging in this promising technology, or referring to Section 2.3

The sentence was removed from the paragraph. We provided further explanation in section 4.

  1. Challenges and Limitations of Using EVs for Therapeutic Purposes in Neuroregeneration

In vitro and in vivo studies of EVs have demonstrated their therapeutic efficacy and potential in nerve regeneration, especially in the CNS, which has limited nerve regeneration capacity and may be a source of hope for currently incurable nervous system diseases. Although SC-EVs use in nervous system injury and diseases is still quite limited, the interest in this field has increased. Despite some studies showing therapeutic efficacy, there are no approved and clinically applied treatment methods yet. We believe that the lack of an approved treatment is due to several problems that need to be overcome. Firstly, there are some disadvantages to the use of SC-derived EVs in treatment, including:

  1. SC-EVs have short half-lives,
  2. SC-EVs have limited targeting capacity,
  3. SC-EVs may be insufficient in content (DNA, miRNA, proteins etc.),
  4. BBB transition not yet fully proven,
  5. Accumulation in the lung, liver, and spleen has been reported [234, 235]

 In addition to these disadvantages, different problems arise depending on the source from which the EVs are obtained. In addition to these limitations, the use of EVs isolated in vitro from stem cells, especially for patients with CNS diseases, suffers from donor heterogeneity and is one of the most important problems in the application of MSCs and EV therapies in the clinical setting [236]. Moreover, other problems must be overcome in the clinical use of EVs. These challenges include:

1- Standardisation: Establishing standardised protocols for the isolation and characterisation of EVs to ensure consistent quality and reproducibility of therapeutic preparations is crucial. Different populations and subpopulations of EVs need to be isolated according to their biophysical and biochemical properties. However, current separation methods are complex, and further studies are needed, especially regarding EVs isolated from cell culture media [237, 238]. Developing a standard protocol for EVs to be obtained from stem cells is necessary to pave the way for their use in medicine. The efficiency of EVs obtained from sequestrants collected from different MSCs from cell culture also varies. Even the isolation method plays a critical role in EV efficiency. For example, various methods have been used for the isolation and purification of EVs, including:

  • Differential Centrifugation/Ultracentrifugation with or without Sucrose Gradient Cushion,
  • Polymeric Precipitation Isolation,
  • Size Exclusion Chromatography,
  • Immunoaffinity Isolation,
  • Ultrafiltration And Microfiltration Technologies,
  • Microfluidic Devices
  • EVs Isolation Reagents [14, 15, 239, 240, 241].

 Among these methods, the most frequently investigated method is ultracentrifugation. This method allows isolation from large-scale samples, especially in cell culture. However, in recent years, there has been evidence that high-speed centrifugation (100,000 x g) may affect EV structure and function [242]. Ultracentrifugation is also a costly method in terms of equipment, but sucrose density gradients can also be used in centrifugation-based methods to improve isolation. However, a disadvantage of this method is the need for additional purification to separate EV subpopulations from other microparticles with similar densities and the density gradient matrix [243]. Combined methods may provide higher yields while maintaining EVs' biophysical and protective properties, often resulting in more successful isolation, especially from stem cell cultures [244]. Bringing together relevant standardisation studies and establishing standard protocols to ensure that the sample obtained in EV isolation has a minimum contamination rate is important in the clinical approval and implementation phase. Another important element for standardisation is the need to characterise isolated EVs and establish protocols at this stage. Although the characterisation of the sample obtained after isolation, concentration, and size analysis can be performed, detailed characterisation of the molecular composition of EVs and EV subpopulations is still a challenge [245]. Protocols used to measure size and concentration have primarily been established. Further molecular analysis protocols are also needed to examine the efficiency of EV isolation in terms of the degree of purity and the mechanisms that may affect the target tissue, biodistribution, and regeneration. Particular challenges for the characterisation of EVs are low specificity and sensitivity.

 2-Storage Optimisation: Optimal storage conditions for SC-EVs must be determined to maintain their stability and therapeutic efficacy during storage and transportation. Evidence suggests that -80°C is the optimal temperature to preserve EV content for downstream molecular profiling. This temperature has shown its usefulness in protecting EVs, as it is effective in long-term storage. However, refreezing of thawed EVs for use and the continuation of this cycle has been found to cause significant losses in the structure and content of EVs. Therefore, the importance of avoiding a few freeze-thaw cycles needs to be emphasised [246].

 3- Safety of SC donors and isolated EVs: Here, it is also necessary to pay attention to the donors of MSCs at the source of their acquisition, especially in cases where cultured EV isolation is performed. To standardise MSC recovery, the effect of variability in MSC donor characteristics during production on the isolation process should also be considered [247]. Recent studies have found that EVs carry the characteristics of cells from young or old donors and transmit them to target cells. In this case, epigenetic modification of genes and the composition of miRs associated with SC proliferation, self-renewal, and differentiation patterns can be observed. Changes in target cells are thought to be caused by either mRNA transcripts or miRs transferred by EVs. However, it has been emphasised that other cargoes (such as proteins and metabolites) within the EV may also be involved in this process [248]. For this reason, the trophic structure and cytokine profiles of MSCs may depend on the donor's individuality and directly affect the therapeutic efficacy of the EVs isolated from them. Good Manufacturing Practice (GMP) conditions and standardisation are required to minimise donor changes and their effects.

In addition, SC-EVs need to be sterilised after the isolation phase and before application. In this regard, EVs are generally separated from possible pathogens by subjecting them to filtration. Ensuring the sterility of EVs without compromising their content and structural integrity is important for safe clinical applications.

 4- Scalable Production: Many studies on SC-EV isolation have generally been performed using cell culture techniques. However, the biggest problem encountered in cell culture techniques, especially in 2D cultures, is that long-term passage to produce sustainable amounts of EVs may cause cells to lose their clonal and differentiation capacity [249]. Although methods have been developed for SC culture techniques to increase efficiency and avoid existing problems, current EV production methods still have lower yields. This hinders the advancement of preclinical and clinical use of EV as a therapeutic [250]. Therefore, developing large-scale production techniques is necessary to meet the increasing demand for SC-EVs in clinical settings. In studies conducted to increase the efficiency of EVs, it has been pointed out that the efficiency is higher in a 3D culture environment [251]. This demonstrates that maintaining the cells in the spheroid structure may release more bioactive molecules into the local environment, maintaining EV efficiency.

In the culture of SCs, several protocols increase the efficiency of EVs. For example, culture under GMP conditions, especially in a serum-free medium, maintains SC morphology, phenotype and viability while triggering an efficient EV release. Therefore, SCs offer an option for large-scale production when GMP conditions are met. In addition, manipulating EV-biogenesis biology allows for manipulating culture conditions and increased EV yields [252, 253].

Although there are still challenges to overcome, the ability to design EVs with recent studies may allow these problems to be solved to some extent. Studies such as increasing the half-life of EVs, surface modifications of EVs for targeting [254, 255], and increasing the content richness of EVs as carrier systems show that SC-derived EVs can be used actively in therapeutic terms. [241]. Considering SC-EVs' efficacy in peripheral nerve injuries, their contribution to nerve regeneration in the CNS cannot be ignored [256].

  1. In section 2.3, points 1 and 3 appear overlapping. Please clarify.

We have reconstructed all section 2.

Reviewer 2 Report (New Reviewer)

Comments and Suggestions for Authors

The present review article attempts to evaluate the current applications of stem cell extracellular vesicles in the treatment of diseases of the central and neuronal systems. However, the manner in which the information is conveyed and the discussion of the facts lack an appropriate analytical framework, which detracts from the scientific value of the article. A more comprehensive critique is provided in the following sections.

I consider it necessary to point out the proposed classification of MISEV2023 into large and small vesicles, given the great heterogeneity of vesicles and the impossibility of isolating isolated populations.

Figure 1 is not informative and requires a detailed description.

The authors write “These non-membranous nanovesicles were identified by asymmetrical flow field-flow fractionation (AF4)”, it is necessary to explain what the vesicle wall is composed of.

In general, the description of the classification of vesicles is chaotic. The principle of a detailed description of a vesicle type and the lack of emphasis on other subtypes of vesicles are unclear.

The authors describe a non-classical classification of stem cells: “Stem cells can be broadly divided into two classes: embryonic stem cells (ESCs) and mesenchymal stromal/stem cells (MSCs).” This classification requires justification. There is a potency-based classification and a source-based classification.

The authors formally describe the sources of MSCs, reduced immunogenicity, immunomodulatory properties, tissue regeneration and repair, and safety profile, and refer to other review articles. What use are such citations when readers can now find the articles themselves in search databases? An analytical analysis of the observed properties of MSCs is required.

It is not clear from the authors’ description what the problem of the "limitations of stem cells” is and why it should be overcome! Previously, the authors characterize MSCs only on the positive side, which is also a biased and selective approach.

In several parts of the review, the above thoughts are repeated, which indicates poor development of the text and narrative plan.

What criteria were used to select the articles listed in Table 1? To ensure a comprehensive and unbiased analysis, it is essential that the authors are transparent about the criteria used for the literature review and analysis. The description of therapeutic mechanisms is done on several levels, both in the description of the general physiological effects and in the molecular mechanisms. Such a mixed description is not acceptable. The reviewer believes that emphasis should be placed on specific molecular mechanisms that are consistent with the composition of the extracellular vesicle cargo. Furthermore, the mechanisms caused by “overexpression of miR-145” are not the natural mechanisms considered by the authors.

In some places the text is not logically coherent, e.g. the transition from a superficial description of the broad spectrum of therapeutic effects of extracellular vesicles and the subsequent registered clinical studies.

The description of damage to the peripheral nervous system is accompanied by a description of the neuroprotective effects of MSCs and their extracellular vesicles in stroke models, which is inconsistent.

Overall, the manuscript presented is more of a collection of abstracts without systematic analysis, highlighting the main problems in the field under investigation.

Author Response

Response to Reviewer 2:

The present review article attempts to evaluate the current applications of stem cell extracellular vesicles in the treatment of diseases of the central and neuronal systems. However, the manner in which the information is conveyed and the discussion of the facts lack an appropriate analytical framework, which detracts from the scientific value of the article. A more comprehensive critique is provided in the following sections.

  1. I consider it necessary to point out the proposed classification of MISEV2023 into large and small vesicles, given the great heterogeneity of vesicles and the impossibility of isolating isolated populations.

We used the classification according to MISEV2023.

  1. Figure 1 is not informative and requires a detailed description.

We replaced Figure 1 with another figure showing the classification of EVs subtypes.

  1. The authors write "These non-membranous nanovesicles were identified by asymmetrical flow field-flow fractionation (AF4)", it is necessary to explain what the vesicle wall is composed of. In general, the description of the classification of vesicles is chaotic. The principle of a detailed description of a vesicle type and the lack of emphasis on other subtypes of vesicles are unclear.

We removed the sentence. We also provided a new detailed paragraph (see below) about the new identified EV subtypes.

In addition to these EV subtypes, in 2021, D' Acunzo et al. identified a distinct EV population following high-resolution density gradient separation of EVs isolated from Down syndrome and diploid control brains. This has led to identifying a third pool of metabolically competent mitochondrial-derived EVs in the EVs subtype. Studies using the ultracentrifugation method to isolate EVs have shown the presence of mitochondrial material in EV pellets centrifuged at 100,000 x g. These vesicles have so far been interpreted as exosomes carrying mitochondrial proteins, lipids and mitochondrial DNA (mtDNA). Although it cannot be ruled out that some mitochondrial components may be present in exosomes, it has been reported that the vesicle was larger than 1 micron in findings with high mitochondrial evidence [37, 38]. Like other novel vesicles, this vesicle is a novel EV subtype, the biogenesis and physiological processes of which need to be elucidated.

  1. The authors describe a non-classical classification of stem cells: "Stem cells can be broadly divided into two classes: embryonic stem cells (ESCs) and mesenchymal stromal/stem cells (MSCs)." This classification requires justification. There is a potency-based classification and a source-based classification.

We reorganised this part by adding new references and listing the MSC advantages and disadvantages compared to ESCs as the EVs source.

  1. The authors formally describe the sources of MSCs, reduced immunogenicity, immunomodulatory properties, tissue regeneration and repair, and safety profile, and refer to other review articles. What use are such citations when readers can now find the articles themselves in search databases? An analytical analysis of the observed properties of MSCs is required.

Thank you for this comment. We reorganised this part of the review paper in section 3 (see sections 3.1, 3.2, 3.3 and 3.4).

  1. It is not clear from the authors' description what the problem of the "limitations of stem cells" is and why it should be overcome! Previously, the authors characterised MSCs only on the positive side, which is also a biased and selective approach.

We corrected this part and described the problem of the "limitations of stem cells".

  1. In several parts of the review, the above thoughts are repeated, which indicates poor development of the text and narrative plan.

We restructured, rewrote the manuscript and deleted the repeated text.

  1. What criteria were used to select the articles listed in Table 1? To ensure a comprehensive and unbiased analysis, it is essential that the authors are transparent about the criteria used for the literature review and analysis. The description of therapeutic mechanisms is done on several levels, both in the description of the general physiological effects and in the molecular mechanisms. Such a mixed description is not acceptable. The reviewer believes that emphasis should be placed on specific molecular mechanisms that are consistent with the composition of the extracellular vesicle cargo. Furthermore, the mechanisms caused by "overexpression of miR-145" are not the natural mechanisms considered by the authors.

We now have deleted Table 1 and corrected this part to be more transparent. miR-145 part was removed from the text.

  1. In some places the text is not logically coherent, e.g. the transition from a superficial description of the broad spectrum of therapeutic effects of extracellular vesicles and the subsequent registered clinical studies.The description of damage to the peripheral nervous system is accompanied by a description of the neuroprotective effects of MSCs and their extracellular vesicles in stroke models, which is inconsistent.

We rewrote this part of the paper to reflect the reviewer's comments.  

  1. Overall, the manuscript presented is more of a collection of abstracts without systematic analysis, highlighting the main problems in the field under investigation.

We performed a systematic analysis to highlight the main problems in sections 3 and 4.

Reviewer 3 Report (New Reviewer)

Comments and Suggestions for Authors

The manuscript "Applications of Stem Cell-Derived Extracellular Vesicles in Nerve Regeneration" offers an insightful review into how stem cell-derived extracellular vesicles (EVs) can aid nerve repair, bypassing the limitations of direct stem cell therapies like immunogenicity and ethical concerns. It delves into EVs' roles in cell communication, their potential in regenerative processes, and the importance of standardizing EV isolation and characterization for clinical use. The review also navigates through the mechanistic pathways by which EVs contribute to nerve regeneration, emphasizing the need for further research to overcome clinical translation challenges, such as scalability and regulatory approvals. Highlighting the promise of EV-based therapies in regenerative medicine, this manuscript calls for more detailed studies to unlock their full therapeutic potential.

1. Expand the literature review to include a wider range of studies on stem cell-derived extracellular vesicles (EVs), focusing on both the diversity of stem cell sources and the variety of nerve injury models used. Highlighting comparative outcomes can provide a broader understanding of the field.

2. The manuscript should delve deeper into the mechanisms by which stem cell-derived EVs contribute to nerve regeneration. This includes discussing the molecular contents of EVs (e.g., proteins, miRNAs) and their specific roles in enhancing neuronal survival, growth, and differentiation.

3. Address the need for standardization in the isolation and characterization of EVs. Discuss current methodologies and the importance of establishing universally accepted criteria to ensure reproducibility and comparability across studies.

4. Discuss the specific challenges faced in translating bench research on stem cell-derived EVs to clinical applications in nerve regeneration. These might include issues related to scalability, purity, storage, and administration methods.

5. Provide a detailed analysis of the safety profile of stem cell-derived EVs, including any reported immunogenicity and the potential for unintended effects, such as tumorigenicity, especially in the context of long-term studies.

6. Offer insights into the regulatory landscape for the clinical use of stem cell-derived EVs. Include discussions on the status of regulatory approvals, guidelines, and the pathway to clinical trials.

7. Highlight recent technological advancements in the engineering of EVs to enhance their therapeutic potential, including methods for loading EVs with therapeutic molecules or targeting ligands to improve specificity to injured nerves.

8. Encourage the inclusion of more quantitative analyses in reviewed studies to strengthen the evidence base for the efficacy of stem cell-derived EVs in nerve regeneration. This may include meta-analyses or systematic reviews.

9. Discuss the potential for personalizing EV therapies based on patient-specific factors or specific types of nerve injuries. This section could explore how EVs from different stem cell sources might be optimized for individual conditions.

10. Compare the effectiveness and practicality of stem cell-derived EVs with other current therapies for nerve regeneration, such as direct stem cell transplantation, growth factor administration, and synthetic nerve conduits.

11. Address the economic aspects of producing and using stem cell-derived EVs for nerve regeneration, including cost-effectiveness, scalability, and the potential impact on healthcare systems.

12. Delve into the ethical considerations of using stem cell-derived products, focusing on EVs and their implications for patient consent, donor anonymity, and the use of embryonic versus adult stem cells.

13. Clearly outline key unanswered questions and propose future research directions that could help fill these gaps. This might include novel EV sources, combination therapies, or innovative delivery mechanisms.

14. Emphasize the importance of considering patient outcomes and quality of life in studies of stem cell-derived EVs for nerve regeneration. Encourage the inclusion of patient-reported outcome measures in future research.

15. Briefly discuss any environmental and sustainability considerations associated with the production and use of stem cell-derived EVs, given the increasing awareness of the ecological impact of biomedical research and therapies.

Comments on the Quality of English Language

Ensure consistent use of specific terms related to stem cell biology and extracellular vesicles throughout the manuscript. Variations in terminology can confuse readers or detract from the clarity of the discussion.

Author Response

Response to Reviewer 3

The manuscript "Applications of Stem Cell-Derived Extracellular Vesicles in Nerve Regeneration" offers an insightful review into how stem cell-derived extracellular vesicles (EVs) can aid nerve repair, bypassing the limitations of direct stem cell therapies like immunogenicity and ethical concerns. It delves into EVs' roles in cell communication, their potential in regenerative processes, and the importance of standardising EV isolation and characterisation for clinical use. The review also navigates through the mechanistic pathways by which EVs contribute to nerve regeneration, emphasising the need for further research to overcome clinical translation challenges, such as scalability and regulatory approvals. Highlighting the promise of EV-based therapies in regenerative medicine, this manuscript calls for more detailed studies to unlock their full therapeutic potential.

  1. Expand the literature review to include a wider range of studies on stem cell-derived extracellular vesicles (EVs), focusing on both the diversity of stem cell sources and the variety of nerve injury models used. Highlighting comparative outcomes can provide a broader understanding of the field.

We expanded the literature review in section 3, focusing on the diversity of stem cell sources and the variety of nerve injury models used. We rewrote section 3 based on the reviewer's recommendation.

  1. The manuscript should delve deeper into the mechanisms by which stem cell-derived EVs contribute to nerve regeneration. This includes discussing the molecular contents of EVs (e.g., proteins, miRNAs) and their specific roles in enhancing neuronal survival, growth, and differentiation.

We added additional information on the mechanism of stem cell-derived EVs and their molecular content roles. These changes can be seen in sections 3.2, 3.3 and 3.4. 

  1. Address the need for standardisation in the isolation and characterisation of EVs. Discuss current methodologies and the importance of establishing universally accepted criteria to ensure reproducibility and comparability across studies.

We addressed the need for standardisation in the isolation and characterisation of EVs in section 4, lines 654-692.

  1. Discuss the specific challenges faced in translating bench research on stem cell-derived EVs to clinical applications in nerve regeneration. These might include issues related to scalability, purity, storage, and administration methods.

We mentioned scalability, purity, storage, and administration methods of stem cell-derived EVs using examples of translational research of EVs in section 3.2.

  1. Provide a detailed analysis of the safety profile of stem cell- derived EVs, including any reported immunogenicity and the potential for unintended effects, such as tumorigenicity, especially in the context of long-term studies.

We mentioned safety profile and we expanded the document as seen below:

Safety profile: MSCs have shown a favourable safety profile in preclinical and clinical studies, with a low incidence of adverse effects. This makes them a promising therapeutic option for many diseases and conditions [93].

Non-tumorigenic nature: MSCs have a lower risk of forming teratomas or other tumours, unlike embryonic stem cells. This characteristic enhances their safety profile and reduces concerns associated with tumorigenicity [93.

MSCs, however, also have disadvantages such as donor site morbidity, variability of reproduction and differentiation capacity, inadequate retention and differentiation of cultured cells, declining intrinsic activity and functionality of obtained cells over the donor's age and inconsistent quality control for large-scale cell production [, 94, 98].

These disadvantages are difficult to overcome, and focus has been placed on alternative cell-free therapies, specifically on using MSC stem cell-derived EVs.It is well-known that stem cells actively employ EVs in cell communication within their mi-croenvironment. [94].

In addition, intercellular interactions mediated by MSC-derived EVs have been shown to play an important role in disease treatment. The bioactive molecules carried by EVs exert their effects on target cells through several mechanisms, including the direct stimulation of target cells via surface-bound ligands, transfer of activated receptors to recipient cells and EVs' epigenetic reprogramming of recipient cells through the delivery of functional proteins, lipids, and non-coding RNAs [99]. Exosomes can engage in communication with both nearby and distant cells. Numerous studies have investigated the therapeutic efficacy of stem cell-derived EVs in various disease models.

  1. Offer insights into the regulatory landscape for the clinical use of stem cell-derived EVs. Include discussions on the status of regulatory approvals, guidelines, and the pathway to clinical trials.

We mentioned clinical studies with stem-cell-derived EVs, Also, we emphasised a MISEV guideline 2023 to explain insights and regulations which are evolving during the production and isolation of EVs.

  1. Highlight recent technological advancements in the engineering of EVs to enhance their therapeutic potential, including methods for loading EVs with therapeutic molecules or targeting ligands to improve specificity to injured nerves.

We added detailed information about methods for loading EVs with therapeutic molecules or targeting ligands to improve the specificity of injured nerves.

  1. Encourage the inclusion of more quantitative analyses in reviewed studies to strengthen the evidence base for the efficacy of stem cell-derived EVs in nerve regeneration. This may include meta-analyses or systematic reviews.

We added some quantitative analysis about the efficiency of EVs, as seen in Figures 2 and 3.

  1. Discuss the potential for personalising EV therapies based on patient-specific factors or specific types of nerve injuries. This section could explore how EVs from different stem cell sources might be optimised for individual conditions.

We mentioned patient-specific factors under conditions with specific types of nerve injuries. This information was added throughout section 3. 

  1. Compare the effectiveness and practicality of stem cell- derived EVs with other current therapies for nerve regeneration, such as direct stem cell transplantation, growth factor administration, and synthetic nerve conduits.

Stem cell transplantation information is added in section 3.

The therapeutic potential of stem cells and their underlying mechanisms largely depend on the conduction pathways and timing of transplantation, particularly in the context of peripheral nerve injuries, one of the most effective methods [108]. Numerous studies have demonstrated that systemic administration of stem cells in experimental stroke models reduces post-stroke brain damage, enhances neurological recovery, and activates neurodegenerative processes [109].

  1. Address the economic aspects of producing and using stem cell-derived EVs for nerve regeneration, including cost- effectiveness, scalability, and the potential impact on healthcare systems.

We addressed aspects of producing and using stem cell-derived EVs for nerve regeneration, including cost-effectiveness, in section 2.3.

  1. Delve into the ethical considerations of using stem cell- derived products, focusing on EVs and their implications for patient consent, donor anonymity, and the use of embryonic versus adult stem cells.

Ethical concerns are mentioned in several places in the document.

  1. Clearly outline key unanswered questions and propose future research directions that could help fill these gaps. This might include novel EV sources, combination therapies, or innovative delivery mechanisms.

We added the following text, lines 530-545 (see below).

To date, no treatments can regenerate axons completely after SCI. However, some progress has been made in understanding the reasons for the failure of CNS axons to regenerate, and this has been translated into preclinical models of SCI that show some promise [176, 177]. For example, research in inflammation, scar formation, cell transplantation, and biomaterials to promote repair have all demonstrated some success in preclinical models of SCI. More recently, combinations of biomaterials, stem cells, growth factors and drugs have led to improvements in axon regeneration over single treatments [178]. For more information on these research areas, the reader is directed to some excellent reviews elsewhere [102, 179-183].

EVs in the CNS may be novel "non-cellular" therapy and are known for their ability to exchange information and play a part in functional activities in the CNS [184]. MSC-derived EVs (MSC-EVs) are rapidly gaining traction in promoting SCI repair, both independently and as combinatorial treatments [185]. In general, EV research in SCI has focused on three main areas: (1) inhibition of inflammation; (2) activation of axon regeneration or reconstruction of damaged circuits to promote functional recovery; and (3) their combination with biomaterials to support growth factor delivery and guided regeneration [186, 187].

  1. Emphasise the importance of considering patient outcomes and quality of life in studies of stem cell-derived EVs for nerve regeneration. Encourage the inclusion of patient-reported outcome measures in future research.

We referred to patient outcomes throughout the document.

  1. Briefly discuss any environmental and sustainability considerations associated with the production and use of stem cell-derived EVs, given the increasing awareness of the ecological impact of biomedical research and therapies.

As a response, we added the paragraph below in section 4.

 4- Scalable Production: Many studies on SC-EV isolation have generally been performed using cell culture techniques. However, the biggest problem encountered in cell culture techniques, especially in 2D cultures, is that long-term passage to produce sustainable amounts of EVs may cause cells to lose their clonal and differentiation capacity [249]. Although methods have been developed for SC culture techniques to increase efficiency and avoid existing problems, current EV production methods still have lower yields. This hinders the advancement of preclinical and clinical use of EV as a therapeutic [250]. Therefore, developing large-scale production techniques is necessary to meet the increasing demand for SC-EVs in clinical settings. In studies conducted to increase the efficiency of EVs, it has been pointed out that the efficiency is higher in a 3D culture environment [251]. This demonstrates that maintaining the cells in the spheroid structure may release more bioactive molecules into the local environment, maintaining EV efficiency.

In the culture of SCs, several protocols increase the efficiency of EVs. For example, culture under GMP conditions, especially in a serum-free medium, maintains SC morphology, phenotype and viability while triggering an efficient EV release. Therefore, SCs offer large-scale production options when GMP conditions are met. In addition, manipulating EV-biogenesis biology allows for manipulating culture conditions and increased EV yields [252, 253].

Round 2

Reviewer 3 Report (New Reviewer)

Comments and Suggestions for Authors

The authors addressed all my concerns, and the current version of the manuscript is acceptable for publication.

Comments on the Quality of English Language

Minor editing of English language required

This manuscript is a resubmission of an earlier submission. The following is a list of the peer review reports and author responses from that submission.

Round 1

Reviewer 1 Report

Comments and Suggestions for Authors

Please revise figure 1 and add figures of all 6 types of EVS. Please also present same EM figures of all 6 types with similar scale.

I suggest draw a shematic figure to present it better: "Exosomes are produced by plasma membrane fusion of multivesicular endosomes (MVEs) followed by the release of intraluminal vesicles (ILVs), while microvesicles are secreted by outward vesiculation of the plasma membrane."

Lin 97 change "Microvesicles and Their Biogenesis." to Microvesicles Biogenesis.

I suggest for both biogenesis section represent 2 figures demonstrate the mechanisms and pathways.

Please change "mesenchymal stem cells" into "mesenchymal stromal/stem cells"

Lines 169-177 is repeated before in lines 145-165

Line 191: Exosomes or EVs?

Some old references can be replaced with new ones (2018 to 2024)

Line 196: it is not clear.

Line 210: please use abbreviation "MSCs"

Line 216: needs a reference.

Section 2.1 and 2.2: Please provide a table to present the results of study better. It is the heart of your article so it needs to be presented better than other sections.

Please write a section to explain challenges and limitations of using EVs for therapeutic purposes in neuro regeneration

Comments on the Quality of English Language

There is some type writing errors such as:

line 56 mcrovesicles

Lines 137, 142 and 181 Repeat of abbreviation and complete name : mesenchymal stem cells (MSCs)

In Table 1: EVs should present in complete form in title and abbreviation in parentheses

Heading of table 1 is not uniformed. the text in the table also needs to edit. Some parts capital letters are not appropriate.

References is not written based on journal instruction.

Reviewer 2 Report

Comments and Suggestions for Authors

Applications of Stem Cell-Derived Extracellular Vesicles in Nerve Regeneration

1.     In the introduction part, EVs are defined into 6 types based on release mechanism and size. Explain each of them briefly and also give information about how they differ from each other. I would suggest to cite Karn et al. 2021 paper https://www.mdpi.com/2227-9059/9/10/1373 and Lin et al. paper 2022 https://www.mdpi.com/1422-0067/23/17/10010, where, they have mentioned types of EVs clearly.    

2.     In this review paper, the image has been taken from another source. It would be more appropriate to create it using an image-making tool.

3.     Create at least one images that can help in understanding the structure and biogenesis of EVs.

4.     There are several typographical errors in the paper, such as in line 56, it would be appropriate to correct "mcrovesicles" and others.

5.     In lines 73, it is mentioned that "In addition to approximately 4400 proteins, 194 lipids, 1639 mRNAs, and 764 miRNAs have been in the exosomes content." Please explain if these quantities of biomolecules are universal in exosomes because they are derived from different cells.

6.     The part on exosomes biogenesis is written with very little information. There is no mention of the various proteins involved in biogenesis, and it needs to be more informative.

7.     After writing "extracellular vesicles (EVs)" once, it would be more appropriate to continue referring to them simply as "EVs" rather than repeatedly writing "extracellular vesicles (EVs)."

8.     In lines 148, 152, and 156, various advantages of MSCs such as reduced immunogenicity, immunomodulatory properties, tissue regeneration, and repair have been mentioned. However, it is appropriate to shed some light on how MSCs achieve these advantages and through what action mechanism. This will make it easier to understand the functions of MSCs.

9.     "Overall, the advantages of MSCs, including their availability, immunomodulatory

Properties, regenerative potential, safety profile, and reduced risk of tumorigenicity, make them a favorable choice for various therapeutic applications" This entire line has been rewritten, which has already been discussed above.

In the entire paper, many old references have been used; it would be better to include recent and updated references.

Comments on the Quality of English Language

Minor English editing is required. 

Reviewer 3 Report

Comments and Suggestions for Authors

The manuscript aims at illustrating the applications of stem cell-derived extracellular vesicles in nerve regeneration. The topic is interesting, but there are several inaccuracies in the introduction to EVs biology that should be corrected. In addition, the introduction to stem cells is very schematic and difficult to follow. The final part is interesting but is also very brief and lacks more critical discussion on drawbacks and pitfalls of the reported studies

A few points

Introduction

Some sentences are unclear. Introduction, line 33 “Today, it is known that the structures that provide encoded cellular functions are EVs.” What do authors mean?

Line 39 The sentence “…and small RNA content consisting of the lipid bilayer”. How can small RNA consist of the lipid bilayer?”

The sentence “These are, nanovesicles [11], exosomes [12], microvesicles [13], apoptotic bodies [14], large 42 oncosomes [15], and giant plasma membrane vesicles [16]” proposes a classification of EVs which is not exhaustive but suggests that is exhaustive. This is not correct. The point of the increasing complexity of the EVs should be introduced more correctly.

Section 1.1

The sentence “Exosomes, formed due to the differentiation of the endosome, originate from the cell membrane [17, 18].” is difficult to catch, as the origin of plasma membrane is typical of microvesicles. Could authors explain what do they mean? Otherwise, the sentence appears incorrect

The sentence “In addition to approx. imately 4400 proteins, 194 lipids, 1639 mRNAs, and 764 miRNAs have been in the exosome content.” need both reference and a context, as is possibly related to a database content, and the EV databases should be introduced in the text to readers.

Authors wrote that “The biogenesis of exosomes…begins with the activation of cell-specific receptors”. This sentence appears incorrect. Please specify.

The sentence “These pathways cannot be independent from each other [24, 25]” should be specified better. The fact that exosome release cannot be completely abolished is interpreted as the evidence of co-existence in the same cells of different pathways, not that they are not independent, as it is written below

2. Stem Cell-Based EVs

Please in mentioning “the remarkable ability to self-renew through division and differentiate”, please explain the concept of asymmetric division, in order to make more precise the description

Lines 145-190 The main advantages of MSCs should be summarized in a Table. Besides, a review implicates critical discussion on the state of the art, so disadvantages should be also discussed.

Table 1 introduces the “Use of different stem cell derived EVs in preclinical studies”. However, preclinical studies studies are many more than those listed. Authors should explain the inclusion criteria used to build the table

In section 2.2. Stem Cell-Based EVs in Central Nervous System Regeneration, the administration route in all the studies reported should be mentioned introduced